# An Advanced First Aid System Based on an Unmanned Aerial Vehicles and a Wireless Body Area Sensor Network for Elderly Persons in Outdoor Environments

**DOI:** 10.3390/s19132955

**Published:** 2019-07-04

**Authors:** Saif Saad Fakhrulddin, Sadik Kamel Gharghan, Ali Al-Naji, Javaan Chahl

**Affiliations:** 1Department of Medical Instrumentation Techniques Engineering, Electrical Engineering Technical College, Middle Technical University, Baghdad, Iraq; saif_hm87@yahoo.com (S.S.F.); ali_abdulelah_noori.al-naji@mymail.unisa.edu.au (A.A.-N.); 2College of Dentistry, University of Mosul, Mosul, Iraq; 3School of Engineering, University of South Australia, Mawson Lakes, SA 5095, Australia; Javaan.Chahl@unisa.edu.au; 4Joint and Operations Analysis Division, Defence Science and Technology Group, Melbourne, VIC 3207, Australia

**Keywords:** algorithm, Arduino microcontroller, drone, fall detection, first aid, GPS, GSM, heart rate, smartphone, UAV, WBSN

## Abstract

For elderly persons, a fall can cause serious injuries such as a hip fracture or head injury. Here, an advanced first aid system is proposed for monitoring elderly patients with heart conditions that puts them at risk of falling and for providing first aid supplies using an unmanned aerial vehicle. A hybridized fall detection algorithm (FDB-HRT) is proposed based on a combination of acceleration and a heart rate threshold. Five volunteers were invited to evaluate the performance of the heartbeat sensor relative to a benchmark device, and the extracted data was validated using statistical analysis. In addition, the accuracy of fall detections and the recorded locations of fall incidents were validated. The proposed FDB-HRT algorithm was 99.16% and 99.2% accurate with regard to heart rate measurement and fall detection, respectively. In addition, the geolocation error of patient fall incidents based on a GPS module was evaluated by mean absolute error analysis for 17 different locations in three cities in Iraq. Mean absolute error was 1.08 × 10^−5^° and 2.01 × 10^−5^° for latitude and longitude data relative to data from the GPS Benchmark system. In addition, the results revealed that in urban areas, the UAV succeeded in all missions and arrived at the patient’s locations before the ambulance, with an average time savings of 105 s. Moreover, a time saving of 31.81% was achieved when using the UAV to transport a first aid kit to the patient compared to an ambulance. As a result, we can conclude that when compared to delivering first aid via ambulance, our design greatly reduces delivery time. The proposed advanced first aid system outperformed previous systems presented in the literature in terms of accuracy of heart rate measurement, fall detection, and information messages and UAV arrival time.

## 1. Introduction

Falls among elderly persons 65 years of age and older have gradually increased in recent years according to the American Center for Disease Control and Prevention (CDC) [1]. They found that more than one million elderly people fall and are treated in emergency departments due to a fall that causes a head injury or hip fracture each year in the US [2]. The CDC researchers also found that the rate of fall-related deaths in the US have increased by 30% every year [3]. If this rate continues to increase, seven fall-related deaths in the U.S. can be anticipated every hour by 2030. Risk factors for falls among the elderly include cardiac disease, loss of balance, vision problems, foot pain, Parkinson’s disease and side effects of medication. The more risk factors a person has, the greater their chances of falling. One study in 2015 found a relationship between patient falls and irregular heartbeat [4]. They found that elderly people who suffer a fall are twice as likely to have a common type of irregular heartbeat known as atrial fibrillation (AF). They note that “AF can impair the ability of the heart to pump blood around the body, including the brain and can lead to a reduction in the amount of oxygen going to the brain, causing either a faint or black-out (syncope), or dizziness resulting in a fall in a person who is already unstable”. Falls among elderly people and the need to detect them have increasingly become a research subject of interest. Researchers have designed wearable devices for patients to monitor vital signs (e.g., heart rate (HR), body temperature, and acceleration) and then send a notification to a call emergency center (CEC) when abnormal vital signs are measured or a fall is detected (FD) [5]. When a case is critical, the CEC will then provide first aid to the patient using an ambulance or unmanned aerial vehicle (UAV) [6].

Previous work, reviewed in Section 2, has presented some limitations related to FD and UAV devices. In term of the FD device, related studies have encountered challenges in the accuracy of HR measurement and fall detection because the algorithms that were adopted for this purpose have intrinsically low accuracy. In addition, previous works have not considered the relationship between HR and falling when adopting the algorithm to detect the fall. Finally, some related works have adopted a wireless protocol such as Wi-Fi, ZigBee, or Bluetooth. Because these technologies have short transmission distances, the movement of the patient during daily life activities is limited to areas near to wireless access points. With regard to UAV-related limitations, some previous studies have not considered the geolocation accuracy of the UAV related to the location of the patient on the map. In addition, some of the algorithms adopted for path planning can consume a significant amount of time to plan the path for the UAV, causing an increase in the total time required to transport first aid to the patient. These obstacles motivated us to design and develop a system for providing first aid to patients based on FD and UAV devices, with high HR measurement and fall detection accuracy, high geolocation accuracy, and faster first aid delivery.

Based on previous work, the most useful vital sign monitoring sensors for detecting falls are heartbeat (HB), accelerometer (ACC), Spo2, and ECG sensors [7,8,9,10,11,12,13,14,15,16,17,18,19,20]. HB and ACC, have been found to be the most significant sensors, and so they have been adopted in this work. In addition, the algorithms most often adopted to detect falling in previous work include threshold-based [7], Support Vector Machine (SVM) [8], and Hidden Markov Module (HMM) [9] algorithms. In this paper, a new hybrid algorithm is proposed that combines HR and acceleration measurements to predict falling. This approach also adopts wireless technology with a long transmission distance, specifically a GSM module, which enables the patient to move freely outdoors. Using GSM, patients at risk of falling can be monitored everywhere and at any time. In addition, previous work has presented different algorithms for planning the path of a UAV, such as genetic algorithms and back propagation artificial neural networks. Here, an advanced Smartphone-based program is adopted that uses an intelligent autopilot program and which contained a waypoint mode for planning the path.

In this study, the measurement accuracies of the HB sensor and GPS module were validated relative to benchmark systems. The validation of HR accuracy of the proposed device was performed using statistical analyses such as mean absolute error (MAE) and histogram. The geolocation accuracy of the adopted GPS module was validated relative to that of a consumer-ready device using statistical analysis such as absolute error [10]. In addition, the proposed fall detection algorithm was validated in this study, including the classification accuracy for four kinds of falling and four kinds of normal activity [11]. Finally, this work compared the response times of the system’s UAV to those of an ambulance [12].

An advanced first aid system (AFAS) for the elderly in outdoor settings, based on a wireless body-area sensor network (WBSN) [11], is presented here. The AFAS was designed to monitor and provide first aid to patients prone to falling (such as those with atrial fibrillation). A hybridized algorithm that combines HR measurement and acceleration to detect falls, named “fall detection based on heart rate threshold” (FDB-HRT), is proposed. The FDB-HRT can be used to detect falling by elderly people. The AFAS consists of two main parts. The first part, the prototype fall detection device (FDD), is designed for monitoring HR and detecting falls. It consists of a microcontroller; two bio-sensors (HB and ACC) for HR and acceleration, respectively [13]; a GPS module to track location, and a GSM module to send a notification message to the Smartphone of caregivers at a call emergency center (CEC). The second part [14] is the CEC’s provision of first aid to the patient; it includes a first aid package, a smartphone, and a UAV to deliver the first aid package. The smartphone at the CEC is used for two purposes: to receive messages from the FDD and to plan the path for the UAV [15]. The main principle of the proposed AFAS is the FDD, which is attached to the patient’s upper arm and performs monitoring and decision-making activities according to the FDB-HRT algorithm. Once the proposed FDD detects a body fall and the HR measured is abnormal, the FDD sends messages to the CEC that include the patient’s information (ID, health status, and location). The caregivers in the CEC receive messages to display on the LCD of the Smartphone. Accordingly, the first aid package will be prepared according to the patient status and sent via the UAV to the patient on the basis of the coordinates received in the message.

The contributions of this paper can be summarized as follows:(i)An advanced first aid system (AFAS) for elderly people in outdoor settings, based on a fall detection device (FDD) and an unmanned aerial vehicle (UAV), was practically implemented.(ii)A new proposed algorithm, called fall detection based on heart rate threshold (FDB-HRT), was presented to improve fall detection for elderly patients.(iii)The geolocation error of the fallen patients was improved based on advanced GPS using eight satellites for geolocation.(iv)Heart rate, fall detection, and GPS measurement accuracy were confirmed based on a statistical analysis.(v)Compared to delivery via ambulance, the UAV-based first aid kit reduces delivery time to patients.(vi)The results of this work outperformed similar previous research in terms of heart rate measurement accuracy, fall detection accuracy, UAV time savings and mission success.

## 2. Related Work

Previous studies have presented wearable FD systems used to detect patient falls. Other related works have designed or adopted a UAV to transport a first aid kit to a patient who has fallen or has a critical health issue. This section will consider previous work related to FD and UAV systems separately.

### 2.1. Work Related to FD Systems

The main principle of a FD device is that biomedical sensors called sensor nodes monitor the physiological parameters of the body and send data to a microprocessor or controller. The microprocessor receives data from the sensor nodes, converts the data from analog to digital, codes the digital data, and resends it to the base station through wireless technology. The base station decodes the received data, monitors data values over time, decides whether the data values are abnormal, and informs the CEC to provide first aid when a case is critical. Related works have presented wearable FD devices used for fall detection that adopt algorithms such as threshold-based [7,17,18,19,20], Artificial Neural Network (ANN) [20], SVM [8], and HMM [9] models. The existing literature also includes devices used for FD and monitoring of vital signs parameters such as HR, ECG, and Spo2 [10,21,22,23,24,25,26]. Wireless technology has been adopted by all previous works to serve as the gateway between the wearable device and the base station; these technologies include Bluetooth [8,9,17,23,24], GSM [18], ZigBee [7,21,22,27], and nRF24L01+ [25].

Kańtoch et al. [16] implemented a health monitoring system based on a wearable body sensor network that tracked human physiological signals. The authors investigated the measurements that were collected according to a research scenario during communal daily activities. The system consisted of four sensors (e.g., ECG, temperature, humidity, and ACC), and a Smartphone used as a network gateway to forward the acquired data to a remote medical server. The experimental results indicated that the system achieved 95% accuracy for detecting all activities. The advantages of this system were that it allowed detection of abnormal conditions and supervision during daily activities, such as cardiac rehabilitation. In addition, the new system was a starting point for the development of novel healthcare telemetry services. Wang et al. [7] presented enhanced fall detection based on a WBSN used for monitoring elderly patients for falls. The authors adopted a threshold-based fall algorithm. The system consists of two sensors (HB and ACC), and ZigBee was adopted as the wireless protocol. The results show 97.5% accuracy for detecting falls, with 96.8% and 98.1% sensitivity and specificity, respectively. The advantage of that system was its high detection accuracy. The limitation of the system was a lower probability of detection when prone patients fall from a bed to the ground, a problem that affected the overall accuracy of detection.

Wu et al. [18] developed a novel FD system for elderly patients based on a WBSN. The authors adopted an algorithm for detecting falls based on an acceleration threshold and rotation angle. The system consists of an ACC sensor with a GSM/GPRS module based on SIM908 for the wireless technology. The proposed system exhibited 97.1% sensitivity and 98.3% specificity. The advantage of the proposed system was its highly efficient algorithm. Lu et al. [19] presented a high-sensitivity FD device based on kinematic information about the body. The authors optimized a threshold-based algorithm for the detection of falls and false alarm rate. The system adopted tri-axial ACC and barometric pressure sensors. They adopted ZigBee as the wireless protocol. The results indicated system success, with high FD sensitivity of 91%, a low false alarm rate and low power consumption. Cheng et al. [9] proposed a new technique for daily activity monitoring and FD based on surface electromyography and ACC signals. The authors adopted a histogram negative entropy algorithm to determine static and dynamic active segments. In addition, they adopted double stream HMM to identify dynamic gait activities. The results showed that the daily activity monitoring and FD scheme were performed with a recognition accuracy of greater than 98%. The advantages of this proposed method include low computational costs and the capability to accurately distinguish normal for both dynamic transition activities and falling. A limitation of this FD method was that it might be unable to detect some specific fall types, e.g., fainting from a seated posture or falling from the bed while sleeping.

Finally, Kakria et al. [22] established a health monitoring system for remote cardiac patients based on wireless sensors. The authors proposed a location-based real-time monitoring system comprising a wearable sensor based on WBSNs, a smartphone, and a web interface for monitoring. The system consisted of multiple sensors (HB, blood pressure, and temperature) and adopted Bluetooth as the wireless protocol. The proposed system detected abnormal health measurements such as arrhythmia, hypotension, hypertension, fever, and hypothermia, and sent the monitoring information to the patient’s caregivers in less time than other devices with accuracy of greater than 90%.

### 2.2. Work Related to the UAV System

Previous authors have adopted UAVs for transporting first aid supplies to patients [12,28,29,30]. Other work has focused on path planning for UAVs by adopting back propagation neural networks [30], comparing a genetic algorithm to artificial neural networks [31], and comparing a genetic algorithm to particle swarm optimization [32]. These authors compared algorithms by calculating the time required for the UAV to arrive and comparing that to the arrival time of the emergency medical service (EMS) [33]. Claesson et al. [27] presented a new system based on a UAV for delivering an automated external defibrillator (AED) to the scene of an out-of-hospital cardiac arrest (OHCA). This project was to investigate the feasibility of a UAV system for decreasing response time and delivering an AED to person who is in cardiac shock faster than EMS can do so by ambulance. The authors adopted a geographic information system (GIS) to estimate travel times. They used a location module based on a multi-criteria evaluation model to find suitable placements and visualize response times for the use of a UAV in two different regions, urban and rural. In the urban area, the UAV arrived at the patients before EMS in 32% of cases, with a time savings of 1.5 min. In rural areas, the UAV arrived at the patients before EMS in 93% of cases with a mean time saving of 19 min. The advantages of Claesson et al.’s new system include the potential to reduce the time required to deliver a defibrillator to the scene of an OHCA compared to EMS. The limitations of this new system included that it only used data from cardiac etiology, excluding non-cardiac cases that could have altered the results. In addition, the data on UAV delays are simulations and not directly comparable to EMS response times, as they do not account for the time from call to dispatch or delay in landing procedures.

Dayananda et al. [28] proposed a complete architectural framework for a UAV emergency response system that was capable of partially and fully autonomous administration of an AED. The aim of the project was to minimize travel time to victims of out-of-hospital cardiac arrest events. The authors developed a geographic approach to the placement of a robotic arm on a UAV, equipped with an AED. The system consists of path planning based on GPS, an ultrasonic sensor to detect obstacles, an HB sensor to monitor the heart rate of the person, a LIDAR sensor to maintain the elevation of the UAV, a visual sensor to convert the image to pixels, and a camera to capture the image. From Dayananda et al.’s results, the new system minimized the travel time, and the UAV arrived before EMS. An advantage of that system is that it can monitor and provide emergency response for OHCA.

Pulver et al. [29] developed a new geographic approach to the placement of a network of medical UAVs equipped with AED devices. The aim of the project was to minimize travel time to victims of OHCA. The authors adopted a geographic information system (GIS) to estimate travel times. They used location analysis based on a Maximum Coverage Location Problem algorithm to determine the best configuration of UAVs to increase service coverage within one minute. Those authors found that using existing emergency medical service stations to launch UAVs resulted in 80.1% of cardiac arrest events demanding AEDs being reached within one minute, but using new sites to launch the UAVs resulted in 90.3% of demand being reached within one minute. The new approach presented possible advantages including reduced cost and coverage to minimize the travel time necessary to reach and treat cardiac arrest patients. The new approach has some limitations, such as uncertainties regarding the estimate of potential demand for AED shock therapy, the actual time needed to save the patient, the need for an able-bodied bystander willing to quickly respond with the AED, and obstacles to flying such as buildings. Kumar et al. [12] demonstrated the provision of basic first aid to injured sportsmen in outdoor sports activities, using a UAV as an ambulance. The main purpose of the project was emergency healthcare service. The authors adopted a smart watch for tracking the path. The main components are an autonomous UAV, GPS based on MAX2769, GSM/GPRS using a SIM900 chipset, a ground control station based on Raspberry Pi 3, a camera type gimbal, and fire extinguisher balls. The results indicate that this proposed system is low-cost, shortens the response time for a UAV, has low power consumption because the system uses few electronic components, and provides good medical service.

Wu et al. [30] proposed a novel geolocation error reduction method for a UAV supported by a laser ranging sensor based on ANN. The aim of the project was to improve the accuracy of geolocation for UAVs and to compare the results with the least-squares based method. The authors adopted an artificial neural network based on a back-propagation algorithm. The system consists of a UAV based on a DJI Phantom 1, a laser ranging sensor based on a FLUKE 411D, a microprocessor based on PIC16F877A, a DC to DC converter based on LM2596S, a 3.5 V lithium battery, a microchip based on MPLAB IDE, and Bluetooth communication based on a cellphone. The results indicate that the geolocation error of the UAV using the back-propagation algorithm was 4.35% relative to theoretical value, which is significantly less than the 6.69% obtained with the least-squares-based method. This means that the geolocation accuracy was improved significantly by using the back-propagation algorithm. The proposed system would increase the precision of UAV in complex environments. Gautam et al. [31] proposed a new algorithm for path planning for UAVs to avoid obstacles in their path based on a combination of genetic algorithms and artificial neural networks. The authors adopted the output generated from the genetic algorithms used to train the network of ANN. The system consists of a quadcopter UAV, a computer (2 GB RAM, 2.20 GHz Pentium core2 Duo processor, Windows 7 operating system), a simulation program based on MATLAB R2012a, and a radar sensor for obstacle detection. Those authors found that training neural networks using the output of a genetic algorithm allowed the UAV to plan its path faster and better compared to using a GA alone. The advantage of the proposed system is that it found the best path for the UAV in less time and avoided any obstacles in its path.

Thiels et al. [33] presented a new system for delivering blood products and medications based on UAVs. The aim of this project was to reduce the cost, risk and time involved in the transportation of medical products for emergency cases outside the hospital, which are usually in rural areas. The authors used a quadcopter UAV and packages of medical products. The results were as follows: The system provided the ability to expeditiously transport blood products between centers to resolve shortages without involving humans in the transport process, and the system delivered more than 200 units of blood products in 2013 without any risk. Advantages of this system include its capacity to reduce patient transfers and transportation costs while minimizing hazards to human life. The proposed system has some limitations, including the following: (i) aviation concerns from the crowding effect of proliferating UAVs; (ii) blood products must be packaged in a manner that ensures minimal risks of exposure and tampering during transit; and (iii) protections must be implemented to prevent unauthorized interception of controlled substances.

## 3. System Architecture

As shown in the diagram of the proposed AFAS included in Figure 1, it consists of two nodes (transmitter and receiver). The transmitter node, represented as FDD, is used for monitoring of vital signs (e.g., HR and acceleration) of the body via two sensors (HB and ACC) and making the decision about when the data indicate a critical case. If that occurs, it sends information about the patient (ID, health status, and location) through the GSM/GPRS wireless network to the CEC [34]. The receiver node, represented as CEC, is used to deliver first aid using the UAV. Caregivers at the CEC receive the message from the FDD with information about the patient on the adopted Smartphone. Then, they prepare the first aid kit and send it to the patient’s location using the UAV. In this section, we will describe and explain all parts that can be used for both transmitter and receiver nodes.

### 3.1. Fall Detection Device (FDD)

This work used a prototype fall detection device (FDD) based on WBSN for detecting the falling of the human body. The FDD, as shown in Figure 2a, consists of a microcontroller based on the Arduino pro mini, an HB sensor based on a pulse sensor, a digital ACC sensor based on ADXL345, a GSM module based on SIM800L, a GPS module based on a NEO M8N chipset, and a power supply based on a two lithium-ion batteries (3.7 V/8400 mAh). These parts were assembled onto a PCB circuit and integrated into a lightweight wearable device, as shown in Figure 2b. The FDD monitors the vital sign parameters of HR and acceleration based on the HB and ACC sensors, respectively. The measured parameters from both sensors are sent to the microcontroller for processing and monitoring. In addition, the microcontroller classifies whether a detected measurement is abnormal according to the proposed FDB-HRT algorithm and, if so, sends a message to the CEC based on the GSM module. This message includes patient ID, health status, and the location of the patient, which is provided by the GPS module. All parts adopted in the FDD are described below:

#### 3.1.1. Microcontroller

The microcontroller serves two functions. The first is monitoring and the second is making decisions about whether vital signs are abnormal. These functions were carried out by an Arduino Pro Mini based on the Atmega328P microcontroller. The Arduino Pro Mini consists of 20 pins, which include 14 digital pins and 6 analog pins. In addition, an FTDI cable supplies the Arduino Pro Mini with DC power and connects it to a PC computer through the USB port [35]. The Arduino Pro Mini was programmed using the C++ language, and some off-the-shelf libraries will be used after making some improvements of it to perform the functions described above. HB and ACC sensors are connected to the microcontroller through analog and digital inputs, respectively.

The measurement data, which includes HR and acceleration, are sent to the microcontroller via serial port at a rate of 9600 kbps [36]. The microcontroller processed it and began the monitoring function by monitoring the HR and acceleration of the patient. If the microcontroller detects the patient in a critical case according to the proposed FDB-HRT algorithm built in Atmega328P, it would make a decision according to the patient’s status. The “making decision” function included extracting the patient ID from the Arduino library, getting the measured HR of the patient, and getting the location of the patient on the map through the serial port (RX) of the GPS module at a rate of 9600 kbps. All information collected as part of the decision-making function was composed in a message and sent to the CEC through the GSM module [37].

#### 3.1.2. Biomedical Sensors

Two sensors were adopted in this study and implemented in the FDD prototype for use in measuring the vital signs of HR and acceleration. The HB sensor is based on a pulse sensor and has been considered in some other related works [23,38,39,40,41]. It is a non-invasive sensor, meaning “no insertion in the body”, and is used to determine the heart rate by measuring the variations in the intensity of the light transmitted through capillaries of the blood vessels, which depends on a phenomenon called photoplethysmography [37]. The unit of the heart rate measurement was beats per minute (bpm). The HB sensor consists of a green LED used to illuminate the skin, a photodetector used to absorb the light reflected by blood vessels, an optical filter to eliminate external ambient light of different wavelengths, and a power supply (3–5 V). Finally, this sensor attaches to the body, such as on a fingertip or ear. In this work, the HB sensor was attached to a new location on the upper arm because it is soft tissue, has good conductivity between LED of HB sensor and epidermis, and experiences less movement during normal daily body activities. The second sensor was the ACC; this sensor has been widely considered in related works [19,21,42,43,44] focused on fall detection.

In this work, a digital ADXL345 accelerometer was adopted as the ACC sensor because it is thin, small, and has ultra-low power consumption [44]. In addition, it supports three axes (x, y, and z) with high resolution (13-bit) measurement up to ±16 g, high resolution (3.9 mg/LSB) for dynamic acceleration during motion and formats the digital output data as 16-bit two’s complement [43]. The ADXL345 is programed in C++ using the I2C or SPI library, and it has some built-in functions such as activity, inactivity, tap, and double tap. In addition, the ADXL345 has a fall detection algorithm; this study modified this algorithm to detect falls from all positions and increase the accuracy of fall detection. The ACC sensor measures the gravity (g) relative to the body. The main principle of the ACC sensor is a comparison of the threshold value set for FREE-FALL-INTERRUPT with acceleration magnitude |a| as defined in Equation (1) [17]: (1)$$\left|a\right|=\sqrt{A_{x}^{2}+A_{y}^{2}+A_{z}^{2}}$$
where $A_{x}$, $A_{y}$, and $A_{z}$ represent the acceleration of the three axes x, y, and z. At equilibrium, meaning no movement of the body, the result of |a| equals one because the gravity of each axis is 1, 0, and 0 g, respectively. The benefit of using this equation is that it accounts for acceleration in all directions because any change in the gravity measurement in any direction will indicate body movement. When the body moves, such as when the person is walking, sitting, jumping, or lying down, the value of |a| increases to more than 1 g (note that the movement of sitting down results in a higher value of acceleration than other movements). In addition, when the body falls, the value of |a| decreases to near zero because the acceleration in all axis directions decreases to near 0 g.

#### 3.1.3. GSM Module

Some previous work has adopted GSM for the wireless data technology [45,46,47] in WBSN applications because of the long transmission distances and the need to be used in both indoor and outdoor environments. This work adopted a small GSM module based on SIM800L. It works in the GSM 850 MHz, EGSM 900 MHz, DSC 1800 MHz, and PCS 1900 MHz frequencies [48]. It has 88-pin pads of LGA packaging and comes with all hardware for the interface between this module and the proposed FDD. In addition, the SIM800L was compatible with all device requirements, such as the Smartphone. The SIM800L was connected to a microcontroller and programmed in C++ based on AT commands [49]. It sends/receives data to and from the microcontroller via the serial port at a rate of 4800 kbps. The most significant AT command used in this work was (AT + CMGF = 1), used to connect the SIM chip to the provider network, and AT + CMGS = \” phone number” \, used to send the message to the smartphone of the caregivers. Finally, the GSM module receives all of the required patient information from the microcontroller and sends it as a message to the CEC.

#### 3.1.4. GPS Module

GPS modules have been used for obtaining location information about a patient in medical applications [40,50] as well as other applications, such as car tracking, UAVs, etc. [51,52]. This study adopted an advanced version of a GPS module based on NEO-M8N because it is compatible with the GPS module of the selected UAV, so the location of the patient has less geolocation error and high accuracy for position detection. NEO-M8N is a standalone concurrent global navigation satellite system module. In addition, it provides high sensitivity and low acquisition time while maintaining a low system power requirement. This GPS module has RF optimization, front-end LNA for easier antenna integration, and a front-end SAW filter for increased jamming immunity. The NEO-M8N retains aiding information, such as ephemeris, almanac, rough last position and time, which reduces the time to first fix significantly and improves the acquisition sensitivity [53].

In this work, the GPS module was implemented in the FDD and used for obtaining the patient’s location after a fall occurs. The information obtained by the GPS module includes the latitude and longitude of the patient’s location. After a patient falls, the GPS module starts working and sends the location information to the microcontroller via a serial port at a rate of 115, 200 kbps. The microcontroller then sends the information about the patient’s location to the CEC via the GSM module as a link. This link can easily be opened in the map application of smartphones and does not need a complex program to decode it.

### 3.2. Call Emergency Center (CEC)

The CEC is a dedicated operations center used to provide first aid to patients as soon as possible using the UAV. The CEC has a database of patient information, including the ID and health history of each patient. The patient information data helps the caregivers to decrease the time required to prepare the first aid package because they will send the supplies necessary to help that patient based on their medical history, such as drugs or other medicine. The equipment at the CEC consists of three parts: The Smartphone, first aid supplies, and the UAV. These parts are further described below.

#### 3.2.1. Smartphone

The smartphone is a smart device used for calling, messaging, social media, and other applications. It is compatible with the GSM network and with most types of SIM card. Recently, some related works have adopted a smartphone for WBSN applications for medical monitoring or tracking [8,11,43]. In this work, a smartphone (iPhone 6S) was adopted and located in the CEC with caregivers. It was used for two functions: the first was to receive the SMS message from the FDD, which provides caregivers with information including patient ID, health status, and location. The second function was to plan the path of the UAV to send it to the patient according to the location information received via SMS messages from the FDD. An intelligent software program called Autopilot version 4.7 was installed on the smartphone and used to plan the path of the UAV [54].

It should be noted that the use of smartphones was not considered in FDD, this is owing to a body of research that identifies strong limitations to its effectiveness. For example, when used to monitor patient vital signs, smartphones consume more power than the system proposed here and in the case of heart-rate monitoring, keeping the patient’s thumb aligned to the phone’s screen can be difficult. Additionally, using a smartphone’s accelerometer for fall-detection is impractical for many reasons. Firstly, as fall-detection also relies on obtaining heart-rate measurements, the patient’s movement in their daily life is severely limited to keep to their thumb attached to their smartphone. Secondly, compared to the device proposed here, the fall-detection accuracy of smartphones is relatively low and their sensitivity threshold for fall-detection cannot be adjusted. Moreover, additional sensors cannot be added to the smartphone’s in-built algorithm. In contrast, while currently our fall-detection algorithm is a combination of two medical sensors: heartbeat and acceleration, by utilizing free-license software based on Arduino programming in C++, future development of applications with more sensors is possible. Finally, it should be noted that smartphones are much larger than our device.

As yet, smartwatches have not been adopted to the patient side of FDD for the following reasons: (i) the application algorithms of smartwatches such as HR and acceleration measurements, are difficult to modify and merge into one algorithm, which is how algorithms for fall-detection have been implemented in smartphones, (ii) some smartwatches, especially those used to monitor elderly patients, required a monthly maintenance fee paid to the hospital, and (iii) for location information, smartwatches require another device such as a smartphone, which for the patient means both increased power consumption and device-size.

#### 3.2.2. First Aid Kit

The first aid kit is a collection of medical supplies and equipment used to give medical treatment to a patient who is not near a hospital [55]. There is a wide variation in the contents of first aid kits based on its intended uses and the knowledge and experience of those putting it together and the intended end users. In this study, caregivers will pack the first aid package according to the patient’s health status in the received message in addition to the health history of the patient. The purpose of a first aid kit is to reduce deterioration of a patient’s condition. Therefore, the main aims for delivering first aid kits are (i) to minimize delivery time of the UAV to the patient and (ii) to supply the patient with items they need, for example, a blood pressure monitor, pulse oximeter device, bandages, sterile gauze pads of different sizes and medication (as shown in Figure 3). Instructions inside the first aid pack are also helpful to guide a bystander to apply first aid to a patient before the arrival of a medical team.

#### 3.2.3. Unmanned Aerial Vehicle

Unmanned aerial vehicle (UAV) is a term for an aircraft that does not have a pilot onboard. It is increasingly commonly known as a “drone”. The flight of a UAV may operate with various degrees of autonomy, such as under remote control by a human operator or autonomously via onboard computers [56]. Initially, the Federal Aviation Administration (FAA) of the United States only allowed the use of UAVs for military applications [57]. In February 2015, the FAA moved to allow limited use of UAVs for commercial purposes [57]. In 2016, UAV use in the United States were permitted for commercial and medical applications. In 2018, the UAV regulations advised that operators can fly only during daylight or twilight, with altitude and speed restrictions. Also, UAVs must be kept in the line of sight of an operator.

There are many types of UAV, often classified based on the number of rotors, such as quadcopters, multicopters, and hexacopters. Some previous works have adopted UAVs for transportation of first aid in medical applications [12,28,29,30]. In this work, the UAV adopted to deliver the first aid to the patient was a DJI Phantom 3 Professional quadcopter, shown in Figure 4a [58]. It consists of two anti-clockwise and two clockwise propellors, four brushless motors, four electronic speed controls, an Autopilot microcontroller board, airframe, battery pack (15.2 V/4480 mAh, (Figure 4b)), camera gimbal, landing struts, obstacle detector based on an ultrasonic sensor, remote control, and a GPS positioning system. Remote control of the UAV was compatible with the Smartphone adopted for the caregivers in the CEC.

The caregivers attach the first aid package to the landing struts of the UAV and insert the location information received from the FDD based on the GPS module inside the Smartphone to draw a waypoint path to send the UAV to the patient [59]. In addition, the adopted UAV has some important built-in features, such as a “return to home” option used to autonomously return the UAV to its initial location at the CEC after completing the mission [60], and an intelligent flight battery control system, as shown in Figure 4b, which is used to project the ability of the UAV to deliver the first aid to the patient and return to CEC.

In the present study, an advanced proportional–integral–derivative (PID) algorithm was installed in the Autopilot program, version 4.7, and based on waypoint mode [54], used to draw the flight path of the UAV. Where the traditional approach is to define waypoints in terms of absolute coordinates on a map, the autopilot algorithm defines waypoints in relation to a reference point that allows for moving flight paths of the UAV—Including position, expansion and rotation. The portion of the flight path between two waypoints is called a segment, and the aircraft altitude and speed during each segment is determined by the waypoint settings. Flight paths can be either open or closed, depending on the selected end-point. In addition, the direction and duration of movement along the flight path is determined by the mission type, and some mission types offer mission completion actions. While autopilot is engaged, the UAV’s mission type can still be changed and more than one mission can be executed during a single engagement [54].

## 4. Proposed Algorithm

A fall detection system for elderly people based on a FDD that also provides first aid to them by using a UAV requires the design and implementation of a new algorithm to successfully complete these missions. In this paper, a system for this AFAS is proposed that consists of two algorithms for FDD and the CEC, presented and explained in detail below: 

### 4.1. Fall Detection Algorithm

Threshold-based FD algorithms have been proposed by previous authors [7,17,18,19,20]; those works have encountered some problems such as not proposing the system used for prediction of falling, not considering the relationship between falling and measured HR, and not proposing a system used to monitor the patient after falling, such as their HR, temperature, Spo2, etc. According to a study by the Medrounds Institute [61], the normal HR for the elderly is between 60 and 100 bpm. Any change in the normal sequence of the electrical impulses of the heart can cause an abnormal HR, which is called an arrhythmia. The result of arrhythmias is a HR that is too fast or too slow, termed tachycardia and bradycardia, respectively [62]. Bradycardia is a HR below 50 bpm, and of the two conditions, the elderly are more prone to bradycardia. It may cause symptoms such as fainting, dizziness, light-headedness, falling of the body, and fatigue.

Tachycardia is a HR more than 100 bpm and is most prevalent in 88% of the elderly (those over the age of 70) [62]. Tachycardia may cause falling, shock, pain, anemia, and strong emotion. Because of the importance of the relationship between abnormal HR and falling of the body, in this work, we propose a hybrid algorithm that merges the measured HR and an acceleration threshold to predict falling; we call this the “fall detection based on heart rate threshold” (FDB-HRT) algorithm. The FDB-HRT algorithm was translated to a software program and implemented in the Atmega328P microcontroller in the C++ language. The FDB-HRT algorithm consists of two stages. In the first stage, “preparing and configuration”, the microcontroller defines all variables, sets the configuration of input/output, and sets the threshold values of some parameters. Normal HR (NHR_t_) is set between 60 and 100 bpm. In this work, threshold values for the fall acceleration magnitude threshold (FAM_t_), falling time threshold (FT_t_) and activity acceleration magnitude threshold (AAM_t_) were 0.5 g, 40 ms and 2.5 g, respectively (as shown in Table 1). These values were selected within the range of standard values as presented in [43] and based on several experiments, they provide for precise fall-detection measurement. Moreover, to sense activity in the body, including post-fall, the activity counter threshold (AC_t_) was set at 10 s.

When the microcontroller completes the preparation and configuration stage, it turns on the GSM module to send a message that includes the ID and health history of the patient to the CEC to inform caregivers that the patient is online and using the FDD. After sending that message, the microcontroller turns off the GSM module and transitions to the second stage. This second stage is used to monitor the patient and detect if HR is abnormal and the body is falling. In this stage, the microcontroller turns on the two sensors (HB and ACC) and starts monitoring the patient. The microcontroller checks whether the HR has become abnormal by comparing the value of measured HR (HR_m_) with the set value NHR_t_. If it finds that the HR is less than NHR_t_ or more than NHR_t_ after a time delay of 5 s, the microcontroller will check whether the patient has fallen or not by comparing the value of |a| with the set value of FAM_t_. If the value of |a| is less than 0.3 g and remains that low for a period of time greater than FT_t_, the microcontroller will decide that the patient has fallen with an abnormal HR. It will then turn on the GPS module to indicate the location of patient on the map and turns on the GSM module to send a message that includes “HR abnormal”, “patient fall”, and location information. After a fall, the microcontroller will check whether the patient has returned to normal activity or not by comparing the value of |a| with the set value of AAM_t_ after a delay time of 20 s. In addition, the microcontroller will set an activity counter called (AC_m_); this counter increases when it is detected that the value of |a| is less than the value of AAM_t_.

If the microcontroller finds that AC_m_ is less than AC_t_ over a time period of 10 s, it will decide that the patient is inactive. In addition, it will turn on the GPS module to get the location of the patient on the map and turn on the GSM module to send a message that includes “patient inactive” and location information. Otherwise, if the microcontroller finds a value of AC_m_ more than AC_t_, it will decide that the patient has returned to normal activity and check whether HR is normal after a delay time of 10 s. If it then finds an abnormal HR measurement, the microcontroller will send a message to the CEC including “HR abnormal”, “patient active”, and location information. Otherwise, if the HR has returned to normal, the FDD will return to its regular monitoring, as shown in Figure 5a.

### 4.2. CEC Algorithm

Figure 5b shows a flow chart of the proposed algorithm for the CEC. Caregivers in the CEC will use a smartphone to receive messages from the FFD. They will receive two messages; the first message will be received when the FDD starts monitoring and will include the patient’s ID and health history. The second message is received if the FDD detects a patient fall and includes the health status and location information of the patient. When the patient falls and the FDD sends messages, caregivers will prepare the first aid package according to the health status information received and plan the flight path of the UAV with the Autopilot application in the smartphone, using the latitude and longitude information extracted from the received messages. When finished with the preparation of the first aid package and flight path planning, caregivers will attach the first aid kit to the UAV and send it to the patient according to the selected flight path.

## 5. Experiment Configuration

The performance of the proposed AFAS was evaluated for both FDD and CEC separately, as explained in detail below.

### 5.1. Performance Evaluation of FDD

This section is divided into four experiments of evaluation. The first experiment includes separate performance evaluations for the two sensors (HB and ACC). The second experiment evaluated the performance of the proposed FDB-HRT algorithm. The third experiment evaluated the performance of the GPS module in term of geolocation error. Finally, the transmission of information by the GSM module was tested in experiment five.

#### 5.1.1. Performance of HB and ACC Sensors

In the first experiment, to evaluate the performance of the HB sensor for measurement of the HR, a benchmark (BM) device was considered to validate the measurements of the HB sensor. A Philips MP50 patient monitoring device was adopted as the BM device [63]; it has 99% accuracy for HR measurement. Five healthy adult volunteers (ages 22 to 28 years old) and five elderly volunteers taking medicine (aged 61 to 65), participated in this experiment. The HB sensor of the FDD was attached to the upper arm of the volunteer and the BM device was connected to the fingertip of the same volunteer, as shown in Figure 6. For all volunteers, the measured HR data from both devices were extracted to a personal computer using the software PLX-DAQ V2.0 [64]. In total, 500 samples were collected from each devices (250 from adults and 250 from elderly). PLX-DAQ software was used. The collected HR data shown have a low standard deviation and standard convergence between the FDD and BM device. In addition, the data collected from both devices were examined and used to validate the proposed FDD through statistical analyses such as mean absolute error, and histogram.

In terms of the ACC sensor, five healthy adult volunteers (ages 30 to 35) were invited to evaluate the performance of the ACC sensor at distinguishing between falls and normal physical activity. Four types of fall and four types of normal activity were selected, as shown in Table 2. In this experiment, each volunteer performed each type of fall and normal activity three times to ensure the ability of the adopted sensor to distinguish between each type. One hundred and twenty samples were collected from this experiment. These collected samples were used to validate the accuracy of the ACC sensor at detecting falling by using statistical analyses such as equations for sensitivity and accuracy. The study adhered to the Declaration of Helsinki ethical principles (Finland 1964). Written consent from all volunteers was attained prior to the experiment with a full explanation of the study procedures.

#### 5.1.2. Performance of the FDB-HRT Algorithm

The second experiment evaluated the performance of the proposed FDB-HRT algorithm. This was done by attaching the proposed FDD to the upper arm of three adult volunteers (ages 28 to 31). Each volunteer performed a scenario that combined abnormal HR measurement and falling. The adopted scenario consists of three stages of activity depending on the time required for each stage. In addition, a treadmill was used in an indoor experiment to evaluate the activities of patients, such as standing, running, and falling, as shown in Figure 7.

The indoor experiment was based on a treadmill because the measurement of fall detection and HR were monitored by laptop via a USB cable. In the first stage, the FDD was attached to the upper arm of the volunteer, and the volunteer stood up to start running. In the second stage, the volunteer was asked to run for 3 min so that the HR of the volunteer would increase to more than 100 bpm. Finally, in the third stage, the volunteer was asked to fall after running for 1 min. The total data collected for both HR measurement and acceleration from all stages comprised 300 samples for each volunteer. These data were examined and plotted to distinguish and validate the relationship between abnormal HR and falling of the patient for the proposed algorithm.

#### 5.1.3. Geolocation Error of the GPS Module

The accuracy of location information is very important in this work, so the performance of the adopted GPS module was evaluated via comparison with the GPS coordinates website was installed on a personal computer, which served as the BM device [65]. In this experiment, three cities in Iraq were adopted (Baghdad, Mosul, and Erbil) to test the location information collected from both the GPS module of the FDD and the BM device at the same time. In total, geographic information was collected for 17 locations and validated to test the geolocation error of the proposed FDD using statistical analyses such as absolute error and mean absolute error.

#### 5.1.4. Performance of the GSM Module

The data delivered by the GSM module and received by the smartphone must be accurate and contain all of the information needed [66]. Therefore, the performance of the GSM module of the FDD was evaluated by sending some messages from three different cities in Iraq (Erbil, Mosul, and Baghdad) and checking whether they were received by the smartphone and whether they contained all necessary details.

### 5.2. Time Savings of UAV Relative to Ambulance

Time savings is very important and affects the patient’s health outcome. Here, time savings were represented as the decrease in the time required to deliver first aid to the patient and can be calculated by subtracting the time required for an ambulance to reach the patient’s location from the time required for the UAV to reach the same location, as defined in Equation (2): (2)$$\mathrm{Time}\;\mathrm{savings}={\mathrm{Timespent}}_{\mathrm{ambulance}}-{\mathrm{Timespent}}_{\mathrm{UAV}}$$

In this work, PAR hospital in Erbil, Iraq was adopted as the CEC from which to send the UAV and ambulance. Patient locations were specifically selected for being crowded and difficult to access. Two locations were in a crowded residential neighborhood with narrow streets and about 50 m from a school and two others were in a city center, near a popular market.

The experiment consisted of five steps. First, the caregivers received a message from the FDD on their Smartphone, as shown in Figure 8a. Second, caregivers planned the flight path using the waypoint mode of the Autopilot program, as shown in Figure 8b–d. Third, the UAV flies autonomously from the CEC to the patient according to the selected flight path, as shown in Figure 8e. The time that elapsed to deliver the first aid package by UAV (Figure 9) to the patient’s location was calculated based on the GPS timer. Fourth, the caregivers sent an ambulance (Figure 9) to the patient location and calculated the time required for the ambulance to reach the location using the Smartphone timer. Finally, the average time savings for the four adopted locations was calculated according to Equation (3) as defined below:(3)$$Averagetimesavings=\frac{1}{n}\overset{n}{\underset{i=1}{\sum }}Timesavings_{i}$$
where n represents the number of locations.

## 6. Results and Discussions

The initial results of the experiments described above were encouraging. In this section, the results of each experiment are presented and validated separately.

### 6.1. HR Measurements and Static Analysis Results

Figure 10 shows the HR measurements for the ten volunteers relative to the BM device, respectively. Five hundred samples (250 from adults and 250 from elderly) were collected by the FDD. The HR measured by the FDD for elderly volunteers were closer to the HR measured by the BM than for the adult participants, this was the result of elderly participants taking medicines that stabilized their HR. The data collected from the FDD were validated to determine whether the measurements are accurate enough to provide reliable diagnostic information on the heart health status of the patient. In this work, mean absolute error and histogram [67] were adopted as the statistical analyses to validate the HR measurement.

#### 6.1.1. Mean Absolute Error (MAE)

MAE was adopted to measure the difference in HR measurement between the FDD and BM devices [10]. It is used to determine the difference in values between two variables and can be expressed as in Equation (3):(4)$$MAE=\frac{1}{n}\;\overset{n}{\underset{i=1}{\sum }}\left|estimated\;HR_{i}\;-actual\;HR_{i}\right|$$
where *n* represents the HR measurement samples.

Figure 11 shows that the values of MAE were varied for five volunteers over the range 0.98 to 1.32. For three of the volunteers, shown in Figure 11a,b,e, the MAE was 1.1, 0.98, and 1, respectively. In contrast, the MAE was higher (i.e., 1.3) for the other two volunteers (Figure 11c,d). The mean MAE for HR measurements from the FDD across adult volunteers was 1.14. While the MAE for elderly volunteers was 0.824 (as shown in Figure 12), it should be noted that this value was impacted by both their slower movement compared to the adult participants during the HB sensor experiment, and by the use of medications. The mean MAE for HR measurements from the FDD across all volunteers was acceptably low at 0.982. This result indicates a close agreement between measurements from the proposed FDD and the BM device. In addition, the accuracy of the HB sensor of the proposed device was 99.16%. However, the FDD measurements of HR diverged slightly from those of the BM device, with a standard deviation of 5.983 for the FDD compared to 6.054 for the BM.

#### 6.1.2. Histogram Analysis

A histogram is a plot that allows discovery of the underlying frequency distribution of a set of continuous data [68]. This allows the inspection of the data for its underlying distribution (e.g., normal distribution, outliers, skewness). The data are split into intervals, or bins, each of which is represented by a bar on the plot. The *x*-axis represents the range of data values and the *y*-axis represents the frequency of occurrence of each bin. Shorter bars thus represent fewer data points in a bin, whereas high bars represent bins with more points [69]. Therefore, a histogram was adopted to distinguish whether the HR measurement of the FDD was compatible with that of the BM device or not. Figure 13a shows that the HR data exhibit peaks of 74 and 64 data points in the 80.91 bpm bin of the adult volunteers for the BM and FDD devices, respectively. Figure 13b reveals HR data peaks of 67 and 127 data points among elderly volunteers in the 74.19 bpm bin from the FDD and BM devices, respectively. These results indicate that the data measured by the proposed FDD are comparable with those measured by the BM device.

### 6.2. Fall Detection Accuracy Validation

A total of 120 samples were collected from the experiment to validate the ACC sensor. Table 3 shows the test results for the four types of fall and four types of normal activity. The results indicate excellent performance by the ACC sensor of the proposed FDD, which distinguished between falls and normal activities with an accuracy of 99.2%. In addition, the sensitivity of the proposed FDD for detecting falls was 98.33%. The ACC sensor did not reach the fall detection threshold in one test of activity type F3, in which the patient was afraid of falling backward and used the arm of the chair to break their fall.

### 6.3. Measurements of FDB-HRT Algorithm

Figure 14 shows the experiment results for the validation of the proposed FDB-HRT algorithm. Three hundred samples were collected from each volunteer in real time. Samples were recorded every one second, where the total test time was 300 s for each volunteer. Figure 14a shows that the HR measurement of the first volunteer gradually increased from 109 to 114 bpm and 0.16 g of acceleration when the body fell. In contrast, the HR measurement of the second volunteer, shown in Figure 14b, increased to 107 bpm when the volunteer fell on the ground and reached 0.39 g of acceleration, which is less than the fall detection threshold (0.5 g). The third volunteer, as shown in Figure 14c, recorded an increase in HR and decrease in acceleration to 104 bpm and 0.34 g, respectively, when falling. Overall, the results indicate that the proposed algorithm for the FDD was successful and achieved the objective for which it was built. Therefore, the proposed device can be used for monitoring elderly patients who have a type of irregular heartbeat called atrial fibrillation, which causes falling.

### 6.4. GPS Measurement and Accuracy of Geolocations

Seventeen datasets, which included latitude and longitude information, were collected from 17 fall positions in three cities of Iraq (Baghdad, Mosul, and Erbil). The map website application of a personal computer was adopted as the BM system, and GPS coordinates were collected using the BM system and the GPS module of the FDD at the same time [70]. MAE was employed to determine the difference in measurements between the data collected by the FDD and BM systems. Figure 15 shows the slight differences between the data collected by the two systems. The MAE results for the latitude and longitude data were 1.08 × 10^−5^° and 2.01 × 10^−5^°, respectively, which means that the geolocation error of the GPS module adopted in this paper was very small. In addition, the GPS module was accurate at obtaining the location of the patient when falling.

### 6.5. Testing of Information Delivery

A total of 40 messages were sent from different locations in Iraq via the GSM of the FDD based on a SIM800L board. The messages sent by the GSM module, which contained the patient information (ID, health status, and geographic location) were received on the smartphone at the CEC 100% of the time without any loss of information, as shown in Figure 16.

### 6.6. Time Savings of UAV Relative to Ambulance

After obtaining the patient’s exact position and preparing the flight path for the UAV, the UAV and an ambulance were sent to the four test locations according to the planned paths, and the travel times for the UAV (including flight perpetration) and ambulance were calculated for each site. The time savings for each location and the average time savings across sites were calculated according to Equations (2) and (3). In addition, the travel times of the UAV and ambulance are presented in Table 4. The results show that the UAV reached locations 1 and 2 in 210 s, 90 s before the ambulance, because the street routes to those patient positions were so crowded and included obstacles, resulting in the ambulance taking 300 s to reach the same locations. For locations 3 and 4, the UAV arrived in 240 s compared to the ambulance’s 360 s for a time saving of 120 s because the road to the patient was closed. Overall, the results for these locations in urban areas indicated that the UAV was successful in all missions and arrived at all patient locations before the ambulance with an average time savings of 105 s. This is a clear indication of reduced delivery time of the first aid kit based on UAV, additionally, it reached those locations before the ambulance medical team. This represents a time savings of 31.81% based on using a UAV compared to an ambulance, calculated by applying Equation (5): (5)$$\mathrm{Percentage}\;\mathrm{of}\;\mathrm{time}\;\mathrm{saved}=\frac{{\mathrm{averagetime}}_{\mathrm{ambulance}}-{\mathrm{averagetime}}_{\mathrm{drone}}}{{\mathrm{averagetime}}_{\mathrm{ambulance}}}\times 100\%$$

### 6.7. Battery Life Estimation of the FDD

The life span of electronic components, which includes the bio-sensors, microcontroller, GPS and GSM, refers to the time of the first transmission until those components lost their working capability in the FDD, specially GPS and GSM [45,46,47]. In this study our proposed device turns off the GPS and GSM modules until it detects the patient’s fall, thus reducing FDD current consumption. Battery life can be determined using the equation below Equation (6) [20]:(6)$$Battery_{lifetime}=\frac{C_{battery}}{I_{Total}}$$
where $C_{battery}$ is the capacity of the battery used as a DC power supply, two lithium-ion batteries (3.7 V/8400 mAh) were adopted for this study; *I_Total_* represented the total current consumed by the FDD. Our results indicate that average current consumption of the FDD was 9.54 mA, this was based on a low power-down mode (on/off scheme), and compared with 85.85 mA in traditional operation. Consequently, based on Equation (6), the battery life of the FDD can be extended to 36 days relative to traditional operation (4 days).

## 7. Comparison of Results with Previous Work

Several reliable studies that are similar to the present work in terms of monitoring patient vital signs such as blood pressure, heart rate, fall detection, breathing rate, SpO_2_, patient’s activities, etc., were compared with the results for the FDD. In addition, some related works have adopted algorithms to plan the flight path of a UAV; these were compared with the UAV path planning results for this work to determine the response time of a UAV when used for delivery of the first aid kit. These comparisons can be divided into four sections, as described below.

### 7.1. Comparison of Heart Rate Measurement Accuracy

Previous works have presented heart monitoring devices that use some type of heartbeat sensor, such as a pulse sensor, MAX30100 sensor, etc. In addition, the locations of these sensors are highlighted to explore the best location for such a sensor on the patient’s body. Most previous studies have adopted the fingertip as the heartbeat sensor location, but these locations result in low HR measurement accuracy because they are often not clean and are frequently in motion, causing confusion and inaccuracy of measurement. This study adopted the upper arm as the HR sensor location and obtained a high accuracy of HR measurement. A comparison of the accuracy of HR measurements of the FDD with results from other related works is shown in Figure 17. Obviously, the HR measurement accuracy of the proposed FDD outperformed previous works, with an accuracy of 99.16%.

### 7.2. Comparison of Fall Detection Accuracy

Fall detection accuracy in this study depends on an acceleration threshold, HR measurements, and sensor position. Previous work has presented designs and developed systems for fall detection that use fall detection algorithms including threshold-based, SVM, CHMM, and ANN models [71]. These works do not incorporate HR into fall prediction. In this work, a hybrid algorithm is proposed for FDD that predicts falls based on a combination of an acceleration threshold and HR measurement, called FDB-HRT. The fall algorithm proposed here achieved an overall fall detection accuracy and sensitivity of 99.2% and 98.33%, respectively. The results for the FDB-HRT algorithm are compared with those of similar related works in Figure 18. The adopted FDB-HRT algorithm of the FDD is superior to other fall detection systems in terms of accuracy, as shown in Figure 18.

### 7.3. Comparison of Response Time

Response time is very important when delivering first aid to a patient. Some previous work on the use of a UAV for first aid delivery has focused on adopting an algorithm to plan the flight path of the UAV. Other works have adopted a UAV to deliver first aid without a focus on planning the flight path. In addition, related works have not considered a complete system for fall detection and autonomous UAV-based first aid delivery. This paper adopted an advanced DJI drone and used the Autopilot program on a Smartphone to plan the flight path. In addition, the Smartphone was compatible with the FDD and used to receive information messages from it. The adopted Autopilot program succeeded in shortening the response time for the UAV, with an average time savings of 105 s compared to ambulance response times. In addition, the results for response time can be compared with those of other related works, as shown in Figure 19.

### 7.4. Comparison of Transmission Information Accuracy

The messages received from the FDD on the smartphone were accurate and did not have any loss of information. The GSM module utilized in the FDD achieved 100% message delivery and receipt by the smartphone without any data loss, which is superior to the results observed in related work [71], which had some data loss and a message delivery rate of 93.75%.

## 8. Conclusions

This paper has presented a design and practical implementation of a prototype fall detection device for elderly patients in outdoor environments. In addition, a UAV was adopted to transport first aid supplies to the patient when a fall and abnormal heart rate were detected. The proposed FDD was small, lightweight, and had low power requirements. In addition, we proposed a novel hybrid algorithm for predicting fall by merging a fall acceleration threshold and HR measurement, which we term the FDB-HRT algorithm. The proposed algorithm was validated relative to BM devices using statistical analysis, i.e., mean absolute error and histogram.

The results of validation indicate that the FDD was validated with 99.16% and 99.2% accuracy for HR measurement and fall detection, respectively. In addition, the geolocation error of the GPS module was validated by comparing it with a consumer-ready system, using mean absolute error analysis to examine the data collected from both systems. The adopted GPS module achieved low error relative to the BM system, with 1.08 × 10^−5^° and 2.01 × 10^−5^° of MAE for latitude and longitude, respectively. This work adopted an advanced software program called Autopilot, installed in the Smartphone at the CEC, for planning the flight path of the UAV. Four different test locations were selected as target destinations for the UAV and ambulance, and the response times were calculated for both vehicles. 

The proposed system presented an average time saving of 105 s when using the UAV to deliver the first aid kit to the elderly patient compared with the ambulance response time. The proposed advanced first aid system have exhibit good results, with excellent measurement accuracy in terms of the FDD and excellent response times in terms of sending first aid supplies via UAV. Overall, the proposed AFAS indicates capable results and could be refined and used for monitoring elderly patients and delivering first aid supplies to them as soon as possible. Future work will focus on the development of the whole system, including the design of a system that can receive patient information and send a UAV in response autonomously without human intervention. In addition, the power consumption of the proposed FDD can be investigated to prolong its battery life based on energy-efficient methods such as power reduction techniques or harvesting energy. 

## Figures and Tables

**Figure 1 sensors-19-02955-f001:**
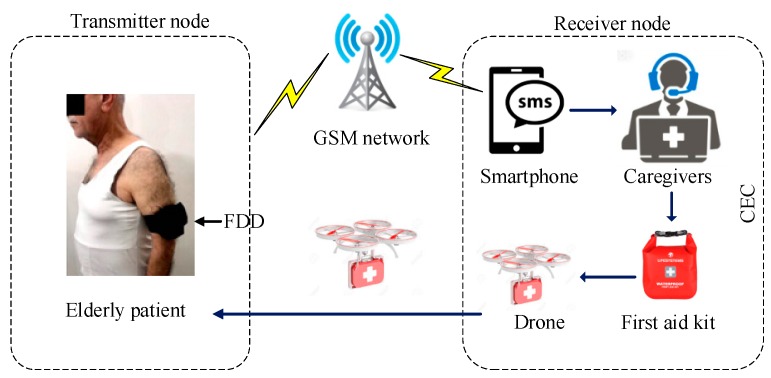
Block diagram of the overall AFAS.

**Figure 2 sensors-19-02955-f002:**
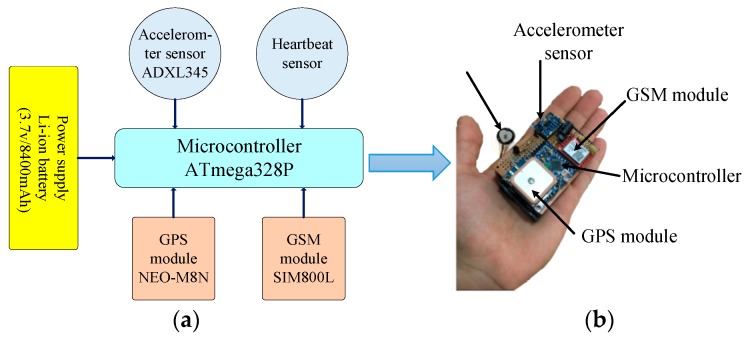
Proposed FDD (**a**) Block diagram, and (**b**) Hardware.

**Figure 3 sensors-19-02955-f003:**
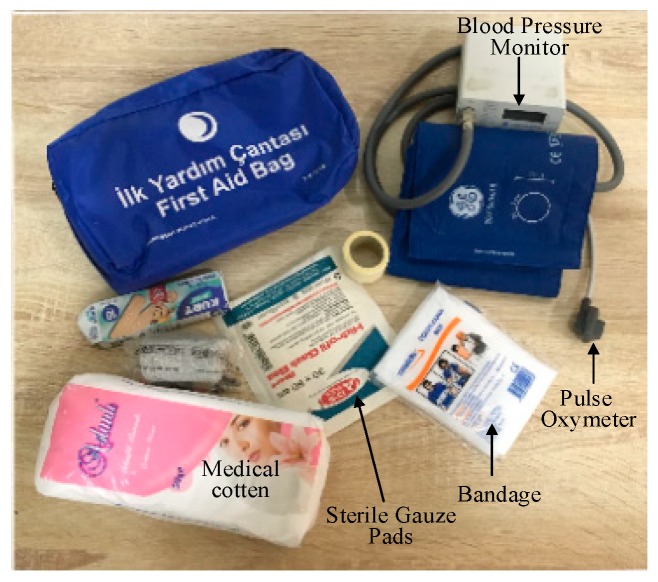
First aid kit with contents.

**Figure 4 sensors-19-02955-f004:**
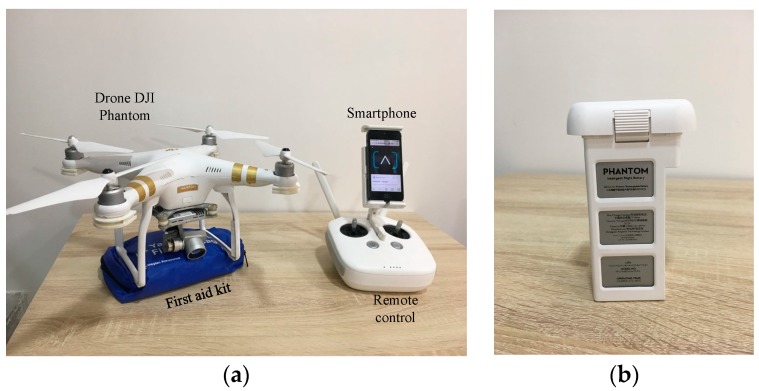
UAV adopted to deliver first aid; (**a**) All components of the DJI Phantom 3, (**b**) Battery pack.

**Figure 5 sensors-19-02955-f005:**
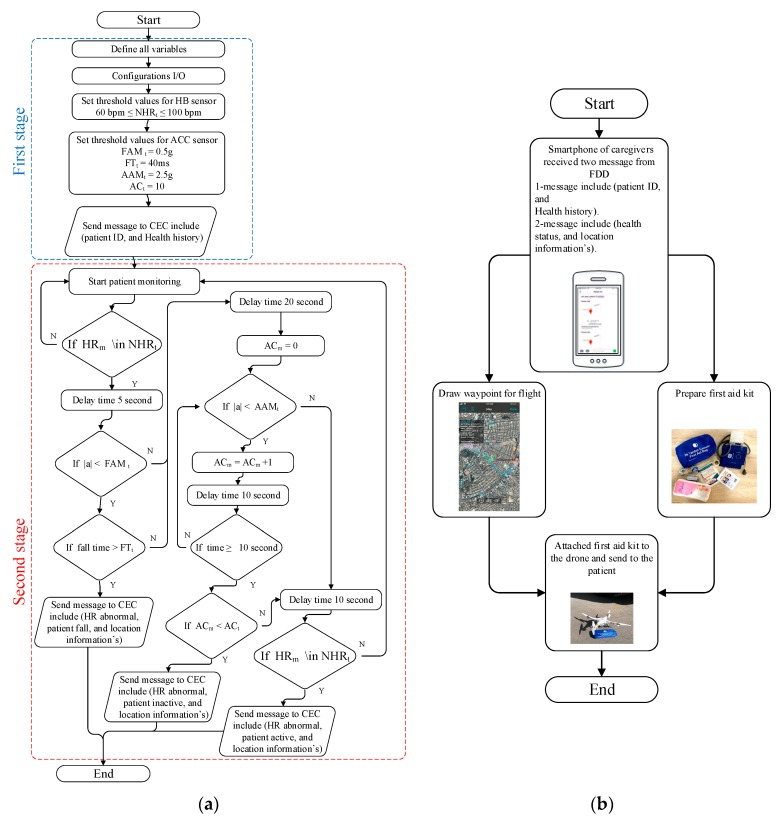
Flowchart algorithms of AFAS for (**a**) FDD and (**b**) CEC.

**Figure 6 sensors-19-02955-f006:**
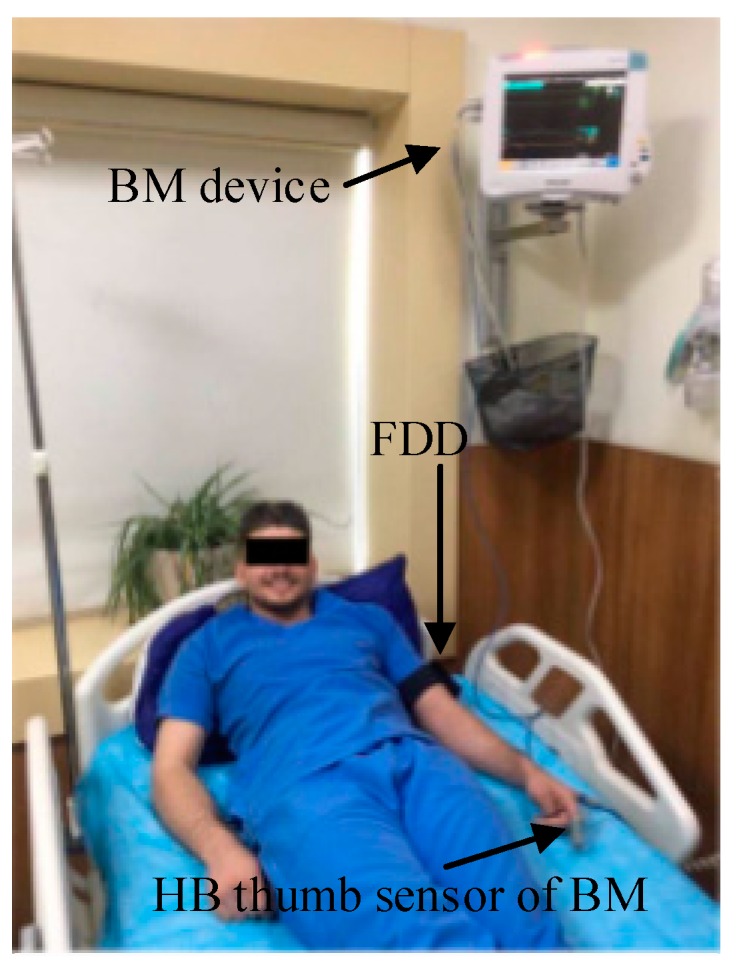
Performance evaluation comparing the FDD and a BM device.

**Figure 7 sensors-19-02955-f007:**
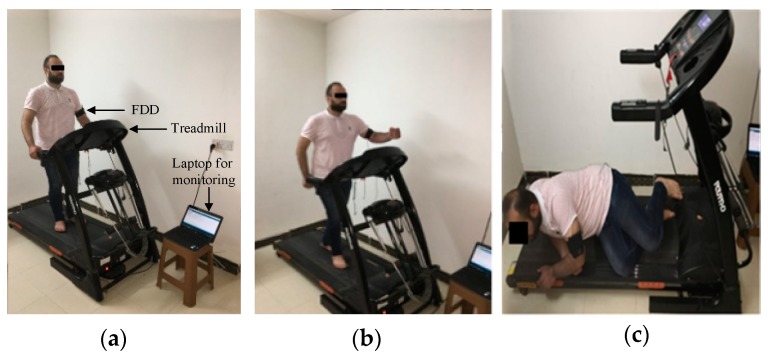
Scenario to evaluate the performance of the FDB-HRT algorithm, with three stages of (**a**) Standing, (**b**) Running, and (**c**) Falling.

**Figure 8 sensors-19-02955-f008:**
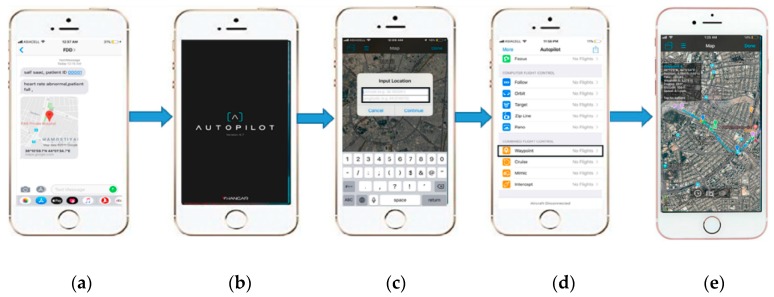
Smartphone of caregivers with (**a**) Patient information (**b**) Autopilot window, (**c**) Location information, (**d**) Waypoint mode, and (**e**) Flight path planning.

**Figure 9 sensors-19-02955-f009:**
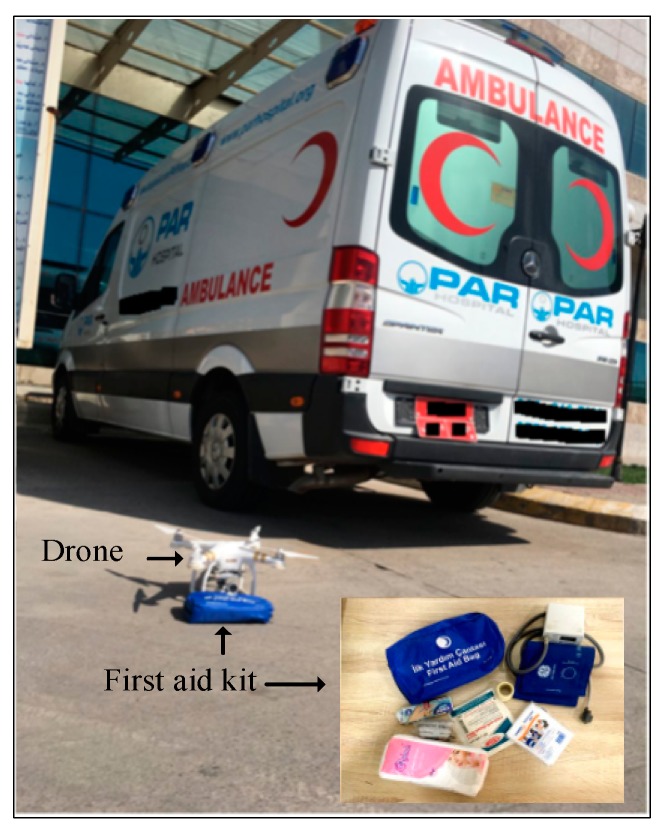
Time response experiment comparing the UAV and ambulance.

**Figure 10 sensors-19-02955-f010:**
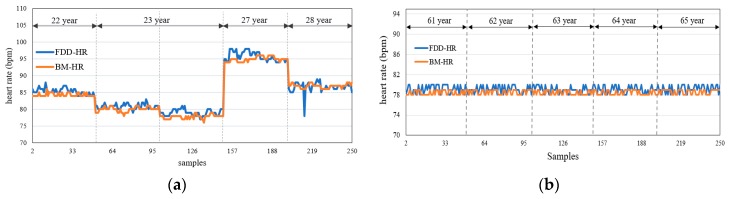
HR measurements for (**a**) five adult volunteers (ages 22 to 28), and (**b**) five elderly volunteers (ages 61 to 65).

**Figure 11 sensors-19-02955-f011:**
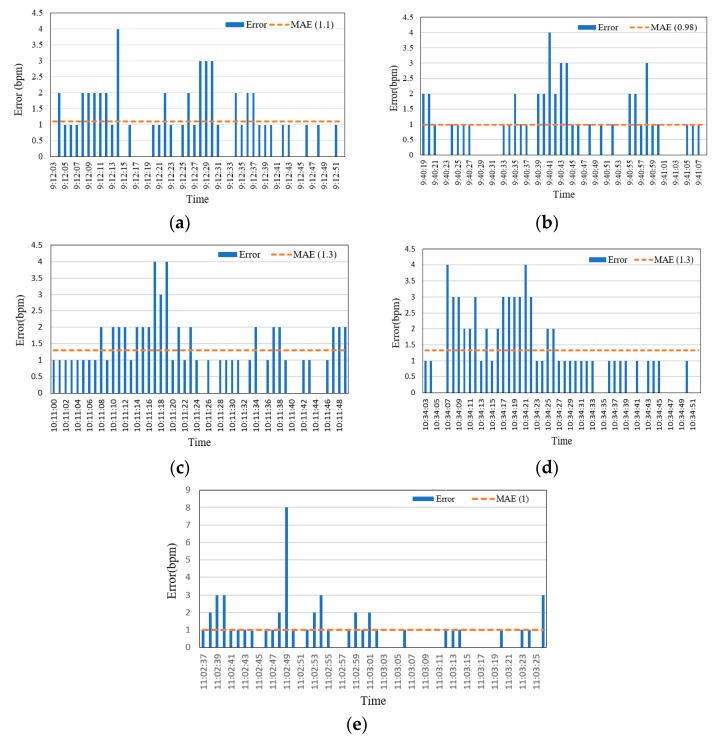
Mean absolute error values for heart rate data from five young volunteers of different ages: (**a**) 22, (**b**) 23 (**c**) 23, (**d**) 27, and (**e**) 28 years old.

**Figure 12 sensors-19-02955-f012:**
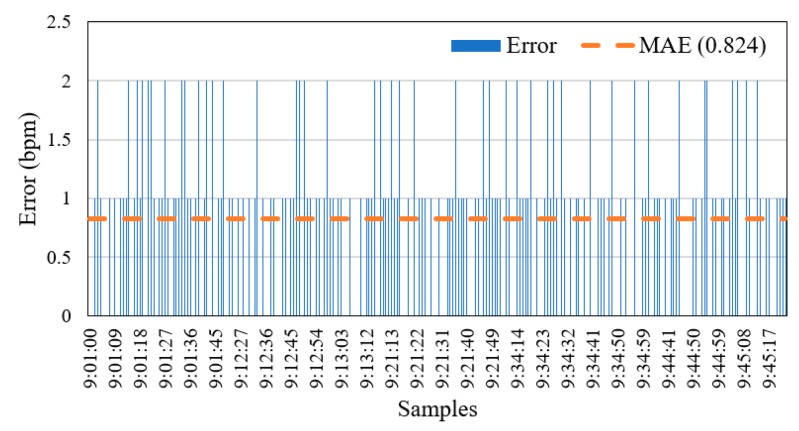
Mean absolute error values for heart rate data from five elderly volunteers, aged 61 to 65 years old.

**Figure 13 sensors-19-02955-f013:**
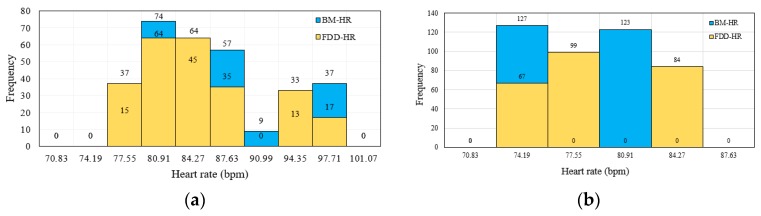
Histogram analysis of HR data measured from both the FDD and BM devices for (**a**) adult volunteers, and (**b**) elderly volunteers.

**Figure 14 sensors-19-02955-f014:**
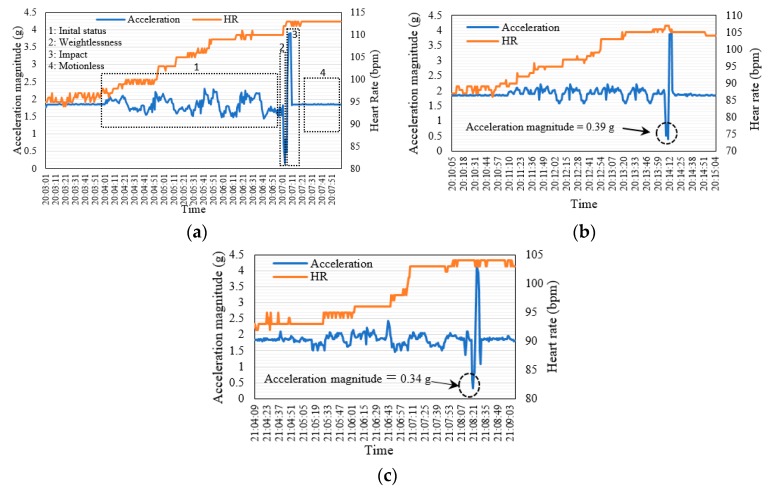
Measurements of the FDB-HRT algorithm for three volunteers: (**a**) 28, (**b**) 30, and (**c**) 31 years old.

**Figure 15 sensors-19-02955-f015:**
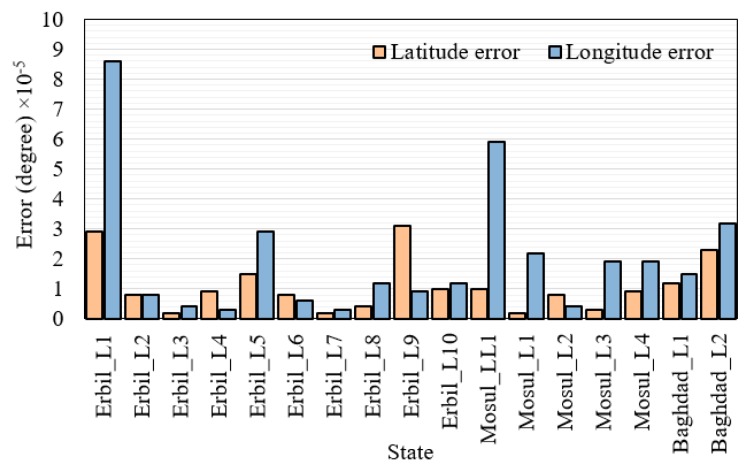
Mean absolute error of the GPS module in terms of latitude and longitude for different locations, (L: location).

**Figure 16 sensors-19-02955-f016:**
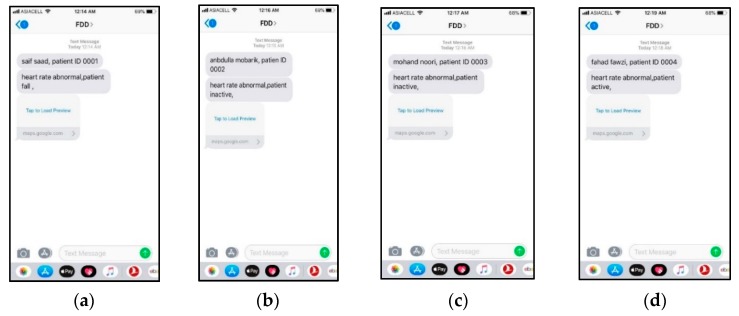
Messages received with details from four different volunteers on the smartphone of the caregivers; (**a**) Message 1, (**b**) Message 2, (**c**) Message 3, and (**d**) Message 4.

**Figure 17 sensors-19-02955-f017:**
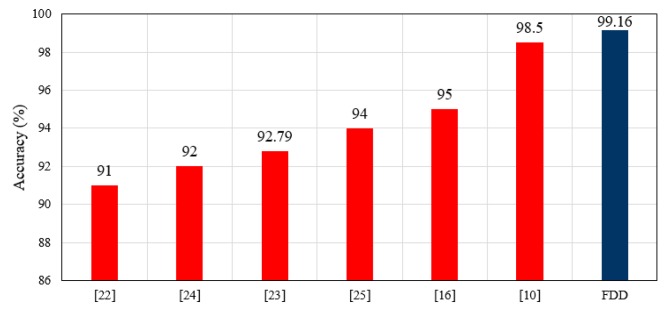
Comparison of HR measurement accuracy between the FDD in this study and the results of related works.

**Figure 18 sensors-19-02955-f018:**
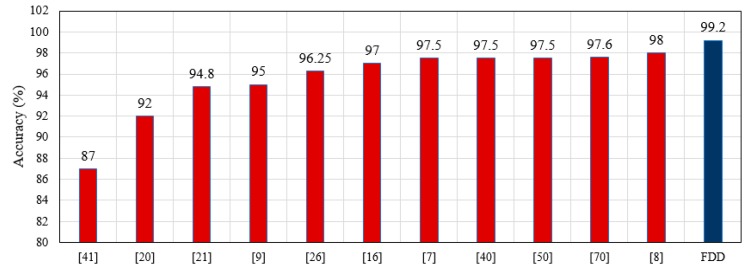
Comparison of the accuracy of the FDD with those of previous works.

**Figure 19 sensors-19-02955-f019:**
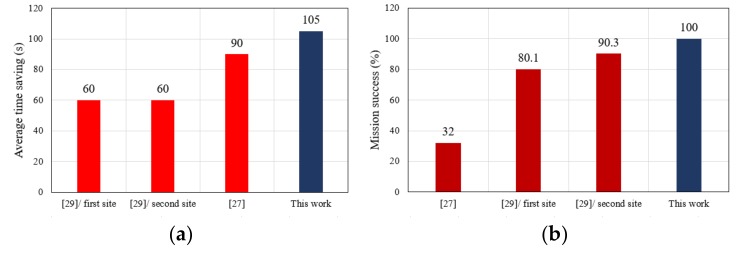
Comparison between the UAV-based system proposed here and other works for (**a**) Time savings and (**b**) Mission success.

**Table 1 sensors-19-02955-t001:** Standard and adopted threshold values of FDB-HRT algorithm.

Parameters	Standard Values [43]	Selected Threshold Values of FDB-HRT Algorithm
FAM_t_	0.313–0.563 (g)	0.5 (g)
FT_t_	20–70 (ms)	40 (ms)
AAM_t_	>2 (g)	2.5 (g)

**Table 2 sensors-19-02955-t002:** Experiments for four kinds of fall and four kinds of normal activities.

Type	Experiment Test
F1	Forward fall, lying on ground
F2	Fall to the right, lying on ground
F3	Backward fall from a seated position on a chair
F4	Forward fall, landing on knees
NA1	Walking
NA2	Ascending stairs
NA3	Descending stairs
NA3	Sitting on chair

F: Fall; NA: Normal Activity.

**Table 3 sensors-19-02955-t003:** Test results for the evaluated types of fall and normal activities.

Type	Experiment Test	Test Result
F1	Forward fall, lying on ground	15/15
F2	Fall to the right, lying on ground	15/15
F3	Backward fall from a seated position on a chair	14/15
F4	Forward fall, landing on knees	15/15
NA1	Walking	15/15
NA2	Ascending stairs	15/15
NA3	Descending stairs	15/15
NA3	Sitting on chair	16/15

F: Fall; NA: Normal Activity.

**Table 4 sensors-19-02955-t004:** Time profile of UAV and ambulance.

Location	Arrival Time of UAV (s)	Arrival Time of Ambulance (s)	Savings Time (s)
Locations 1 and 2	210	300	90 (based on Equation (2))
Locations 3 and 4	240	360	120 (based on Equation (2))
Average time (s)	225	330	
Average time savings = 105 s (based on Equation (3))			
Percentage of time saved = 31.81% (based on Equation (5))			

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
