# Peer review of "An Advanced First Aid System Based on an Unmanned Aerial Vehicles and a Wireless Body Area Sensor Network for Elderly Persons in Outdoor Environments"

_sensors, 2019, doi:10.3390/s19132955_

Reviewer 1 Report

The paper addresses a first aid system prototype able to monitor elderly patients with heart conditions and for providing first aid supplies using an unmanned aerial vehicle. To this extent the authors propose a Fall Detection Device that includes a hybridized fall detection algorithm based on a combination of acceleration and a heart rate threshold.    

The paper is well written and structured, presents a good related work characterization, makes use of an extended and up-to-date set of references and the results of validation seem very promising and, so, it deserves to be published. However, there are some issues that *must* be clarified or solved:  

 A) nothing is mentioned about the aim of delivering the first aid kit. What is the advantage of (just) delivery the first aid kit to near the patient through unmanned vehicle? You have to clarify if you are just trying to solve the delivery “time” problem or anything else because the first aid kit by it self can not help the patient 

 B) in section 4.1 - Fall Detection Algorithm (or in figure 5) the authors specify some constant values such as FAM as 0.5g, FT as 40 ms, AAM as 2.5g, AC as 10, etc. but no reasons for this choice were presented. Why these values? How did you reach those values?  

 C) if the proposed system is targeted to elderly why no elderly was invited for experiments? For instance, would the HB sensor have the same performance and accuracy?  

 D) in my opinion, you should describe better the city sites used for experiments (may be na image, or description about the location of buildings (near or far in meters), buildings height, presence of trees or not, etc. etc.)  

 E) line 695 -  it is not clear if the UAV time includes (or not) the flight preparation  

 Also, in my opinion words likeappeared’, … should be avoided in research papers.

Author Response

Reply to reviewer 1 as in attached file

Reviewer 2 Report

The paper describes a wearable system which detects user falls and hear-rate failures and informs a data center, so that a UAV with a first-aid kit is sent automatically to the user position. The system is properly described, and measurements are reported which support the presented approach. The paper is also clearly written. There are however several issues that need to be clarified in order to justify the presented approach:

- The technologies integrated in the wearable device, such as GPS positioning, cellular connection, accelerometers and a pulse-measurement system, are currently available in conventional smartphones and smartwatches. It seems then that the presented wearable device could be replaced by these devices, so that only an App would need to be developed. Why has this approach not been considered here, while it has been considered in other works such as those detailed in Lines 390-391?

- Time saving results indicate a saving of 90 seconds and 120 seconds with the UAV device, compared to an ambulance. Considering that the UAV brings only a first-aid kit, which in addition must be employed by an injured elderly person, while the ambulance arrives with prepared personnel with full medical equipment, do the measured time saving values justify the proposed system?

- What is the duration of the battery of the wearable device?

Additionally, one minor correction: "ZigBee" instead of "Zigbee" in Lines 65, 145, 156 and 168.

Author Response

Reply to reviewer 2 as in attached file

Reviewer 3 Report

The paper presents a complex solution for first aid to delivery for elderly persons after a fall. The paper represents an excellent work. The topic is highly relevant. It is carefully prepared and easy to follow. It contains a useful literature review on fall detection systems and UAV systems. The architecture of the proposed system is described in details. A new fall detection method is presented. The proposed system is thoroughly evaluated from several aspects and it outperformed the similar existing systems.

I have one concern regarding the practical significance of the proposed system. What happens after the first aid package is delivered? A seriously injured person is unable to use it.

Some remarks:

Fall-detection algorithm: Is NHR_t a scalar or an interval? If it is an interval then the condition should be HR_m \in NHR_t instead of equality check.

Eq (3): the lower index of timesavings is i (not n)

Eq (4): please add index i to the variables within the sum

What kind of algorithm is applied in autopilot program?

References: Autopilot is duplicated in the reference list (15 and 54)

Author Response

Reply to reviewer 3 as in attached file

Round  2

Reviewer 1 Report

It is OK now...